# From Project-Based Health Literacy Data and Measurement to an Integrated System of Analytics and Insights: Enhancing Data-Driven Value Creation in Health-Literate Organizations

**DOI:** 10.3390/ijerph192013210

**Published:** 2022-10-14

**Authors:** Kristine Sørensen

**Affiliations:** Global Health Literacy Academy, Viengevej 100, 8240 Risskov, Denmark; contact@globalhealthliteracyacademy.org

**Keywords:** health literacy, measurement, data governance, analytics, predictions

## Abstract

Health literacy measurement is important to improve equity, health and well-being as part of health system transformation. However, health literacy data of good quality are often lacking or difficult to access for decision-makers. To better inform policy, research and practice, this paper discusses how to move from project-based health literacy data and measurement to an integrated system of analytics and insights enhancing data-driven value creation in health-literate organizations. There is a need for the development of health literacy data pipelines, data dashboards, and data governance mechanisms which are timely and trustworthy. Investing in health literacy data analytics and data governance can pave the way for the integration of health literacy as an acknowledged global health indicator in large-scale surveys, ventures, and daily business. Leadership and management buy-in are needed to steer the process. Lessons learned from decades of measurement research combined with strategic implementation of systematic use of health literacy monitoring may accelerate the progress.

## 1. Introduction

The awareness of the importance of health literacy for equity, health and well-being is moving from the margin to the mainstream as the evidence-base is growing. Subsequently, there is an increased demand for health literacy analytics to inform policy, research, and practice. However, most stakeholders engaging in the promotion of health literacy are still without access to valid health literacy data. The implementation of systemic health literacy data analytics and monitoring mechanisms is still scarce despite the growing amount of health literacy assessment research across the world [1].

Health literacy concerns the knowledge, motivation, and competencies for people to access, understand, appraise, and apply information in everyday life concerning healthcare, disease prevention and health promotion to maintain and promote health during the life course [2]. Health literacy is recognized as a critical social determinant of health [3] and a promising indicator for enhancing economic and social return on investments [4]. However, to inform management and decision-makers a systematic approach is needed to develop effective health literacy data pipelines and dashboards. Through solid measurement, analytics, and predictions, these insights may be integrated into big data analysis for a more comprehensive value creation within and beyond health-literate organizations and systems [5].

Data—combined with analytics—are a valuable asset for any societal system to strengthen management, operational optimization, user insights, personalization, and forecasting [6]. However, systemic approaches to the generation of health literacy insights are still scarce. To fill the gap, health literacy data and analytics can (1) be applied to design an effective data strategy and understand the core value of the data, (2) ensure returns for health literacy data investments, (3) identify the right architecture, technology decisions, and investments required to ensure the ability to support future health literacy data capabilities, (4) apply proper governance in an agile way to ensure the right balance of data access and data security, (5) create a data organization and culture that harness ethics and security, and (6) train the workforce to use the health literacy data as a tool in everyday decision-making [5]. However, health literacy data governance including data availability, usability, consistency, integrity, and security remains an important challenge to accomplish as part of health literacy systems’ capacity [5].

To push the implementation of health literacy data pipelines, dashboards, and analytics to the next level, this paper firstly, highlights the current challenges related to the progression of health literacy analytics from data and measurement to the generation of solid predictions. Secondly, it explores how an organizational foundation for analytical work on health literacy can be put in place in organizations and businesses processes to accelerate the progress on data-driven decision-making and good data governance.

## 2. Health-Literate Organizations Driving Empowerment and Health Equity

Health literacy is a critical determinant of health and a key driver of empowerment and health equity according to the World Health Organization [7]. Nonetheless, health literacy research from across the world suggests that limited health literacy is a public health challenge which differs within and between countries [8]. People with health literacy limitations are at risk of having less knowledge about their health conditions and treatments, poorer overall health status, and higher rates of hospitalization than the average population. Limited health literacy may also impact an individual’s ability to participate in decision-making, follow health recommendations, implement health-promoting behaviors, and engage with preventative health services [9].

Notably, health literacy extends beyond the individual perspective to include a system perspective as well [10,11]. The way an organization responds to health literacy needs is called health literacy responsiveness which can be defined as “the provision of services, programs and information in ways that promote equitable access and engagement, which meet the diverse health literacy needs and preferences of individuals, families and communities, and that support people to participate in decisions regarding their health and social wellbeing” [12]. Health-literate organizations are enabled to shape a culture which promotes equity and inclusiveness, demonstrate effective leadership and management, and ensure robust data collection, monitoring and communications systems and processes are in place. Moreover, health-literate organizations help foster effective communication practices, and have a strong commitment to building the capability of their workforce, engaging meaningfully with the communities they serve, and partnering effectively with other health and social service organizations [12,13].

## 3. From Health Literacy Data and Measurement to Analytics and Predictions

Given the increased awareness toward developing and finetuning relevant health literacy measurement that are sensitive to content, context, and target group; the largest online database of health literacy measures has grown exponentially to contain more than 200 tools. The website, called the Health Literacy Tool Shed (https://healthliteracy.bu.edu, accessed on 13 October 2022) includes information about the measures and their psychometric properties. Promising world-known, measurement tools entail, for example, the Health Literacy Survey Questionnaire targeting population health literacy [8,14] and the Health Literacy Questionnaire [15,16] which is relevant for community intervention design. However, not all measurement tools are freely available, culturally adapted or translated to match requirements for large-scale implementation as part of practice. In most cases, when organizations and businesses consider engaging in health literacy promoting activities and strategies, there are no health literacy data available or the data available may be of poor quality or difficult for decision-makers to access and analyze.

A push is needed to progress health literacy data analytics beyond the status quo moving away from project-based measurement towards an integrated data system with the purpose to generate strong health literacy data which are available and accessible for decision-makers. With time the transformation from ‘co-incidentally’ measurement of health literacy to persistently monitoring the development will allow reliable data analytics and predictions building on data science and quantitative as well as qualitative expertise in order to yield data-driven, value-based recommendations on how to promote and implement health literacy across entities, organizations and business as illustrated in Figure 1.

## 4. Organizational Foundation for Analytical Work on Health Literacy

While a few leading organizations are realizing their success with health literacy measures, most organizations and companies are either still unaware or in an exploration and piloting phase. For progress and upscaling to happen, it is of importance to create a sound organizational foundation for health literacy data analytics. To design a data transformation that delivers value right from the start [6], the creation of a sound organizational foundation require attention to several areas of concern [17]. The areas are adapted for the purpose of this discussion on health literacy data analytics:Define the problem: identify the health literacy gap, problem, or challenge that, with the use of relevant data, will provide more insight to drive the right decisions or provide efficiency on how work can be carried out based on management buy-in and its direction.Problem-solving process: the team working on the defined health literacy problem breaks it down to workable steps to generate and translate the appropriate data to achieve the results.Relevant team of experts: a team including management, data scientists, data engineers, solution architects, and domain experts identifies the right data and works to translate the data to achieve results.Continuous improvement practices: the team embraces fail-fast continuous improvement practices to evaluate their success in translating data to achieve results.Obtaining right data: the team understands and obtains the appropriate health-literacy-related data that explain the health literacy problem to achieve results.Accessible software and hardware infrastructure: the organization secures infrastructure that supports health literacy data processing in a relevant and timely manner.Culture and resources: the organization embraces data insights, sponsors properly resourced teams, and prioritizes analytical development work with regard to health literacy.

A strong foundation for health literacy analytics involves an end-to-end data strategy as much as a strong data team. It is of importance that those involved understand the needs and characteristics of the health literacy data today as well as for future needs.

## 5. Designing a Health Literacy Data Pipeline

Starting with the existing data infrastructure and tools, it is recommended to consider every stage of the health literacy data lifecycle of the business or organization—ingest, store, process, and analyze—and adjust or optimize where needed and will have most impact. At every stage of the lifecycle, the inherent challenges must be addressed to gain the desired outcomes of each phase, while also considering the overall concerns such as data security and governance [17].

Ingest: extracting health literacy data from various sources, detecting what has changed, and move them from the source into a system where they can be stored and analyzed. Thus, there is a focus on data creation, transmission, and ingestion.Prepare: health literacy data come in different structures, sizes, and speeds. Data must be cleansed, normalized, and prepared before they can be stored. Thus, preparation includes data integration and staging.Analyze: transforming the raw health literacy data into actionable information. Data are processed to allow for ad hoc querying and exploration. Thus, this stage includes data inquiry to build models and experiments.Act: actionable outcome of the health literacy data pipeline, health literacy information is translated into insights. Thus, at this stage it is possible to pull actionable insights from the data that were collected, integrated, stored, cleaned, and processed. It might involve in-depth exploration and visualization to better understand the results.

## 6. Good Health Literacy Data Governance

As the health sector becomes increasingly digitized, authorities, providers and other health organizations rely more and more on data to drive their decision-making. As a result, data governance in health services whether public or private; for profit or non-profit, is a particular and growing concern for all stakeholders involved [18]. Data governance concerns the process of managing the availability, usability, integrity, and security of the data in an organization or enterprise. The purpose is to ensure that data quality remains high throughout the lifecycle of the data, so that they are trustworthy, consistent, and not misused [6]. Moreover, any controls that are implemented should be compatible with an organization’s objectives as well as with current rules and regulations such as, e.g., the General Data Protection Regulation (GDPR) within the European Union [19].

## 7. Identifying the Barriers to Health Literacy Analytics Readiness

Health is a knowledge industry, based on data collected to support care, service planning, financing, and knowledge advancement [20]. According to McKinsey [21], a good indication of organizational maturity with regard to data analytics can be seen by how far various data and analytics have penetrated various business units, and the speed with which new project and cases can be implemented. An ability to analyze and mine very large amounts of data, “Big Data”, provides policy and decision makers with new insights into varied aspects of work and information flow and operational business patterns and trends, and drives greater efficiencies, and safer and more effective services in the health field [20]. The health field is one of the fastest-growing areas for data management and data governance, partially because of the enormous amounts of personal consumer, patient, and client information that all health organizations need to deal with. The onus is now on organizations to ensure proper handling and management of these critical data to maintain patient and customer privacy while also enabling full leverage of the incredible capacity it can bring to the users [21].

The Health Catalyst Healthcare Analytics Adoption Model (HAAM) [22] is an example of a tool to evaluate an organization’s progress on becoming data-driven. It has nine levels: (0) Fragmented point solutions, (1) Enterprise data operating system, (2) Standardized vocabulary and patient registries, (3) Automated internal reporting, (4) Automated external reporting, (5) Waste and care variability reduction, (6) Population health management and suggestive analytics, (7) Clinical risk intervention and predictive analytics, (8) Personalized medicine and prescriptive analytics, (9) direct-to-patient or customer analytics and artificial intelligence. Most organizations start at, or near, the bottom of the HAAM. At this point most individual users, teams, and departments are working in silos, implementing fragmented solutions, and working on their own goals while relying on their own data sources. The HAAM model implies how data governance may evolve in nine stages from fragmented point solutions to the use of predictive, prescriptive analytics and artificial intelligence [22].

The journey to prioritize data and become a data-driven health-literate organization starts at the top. Leadership and management buy-in is needed to drive the change, emphasize the importance of integrating the health literacy perspective at every level, and understand that the journey to grow into a data-driven health-literate organization is an ever-evolving state based on the analytic tools and talent available. At an individual level, it entails the skills, knowledge and attitudes of the workforce, the people served, and external partners. At departmental level, it involves the quantity and quality and distribution of resources to generate timely and relevant analytics. At the organizational level, it is about facilitating inter-operationality and coherence.

To become a health-literate, data-driven organization, health systems should invest in an integrative baseline architecture, a dashboard where all health-literacy-related data come together. As organizations and systems progress, they can develop their analytical teams, start utilizing data more efficiently, and finally, drive health literacy improvements within the organization and beyond. At this point, the organizations and systems may become more forward-thinking with data and leverage insights to improve processes, quality, and create predictive models to identify opportunities even closer to the point of interest. Focusing on three areas of concern may leverage an organization to close gaps between delivering insights and acting on them, which will, in turn, accelerate outcomes improvement [22]:Data quality: capture and integrate.Data utilization: grant access and deliver insight.Data literacy: deliver insight and act.

Most organizations using traditional intelligence, will evolve towards more sophisticated AI techniques such as machine learning and deep learning over time Thus, it is essential to keep on top of maintenance and governance of data over time as well by monitoring the entire pipeline and data usage to track trends, bottlenecks, and opportunities [17].

## 8. Conclusions

Health literacy has proven to be an enabler that supports the promotion of equity by improving people’s access to health information and their capacity to use it effectively [23]. In turn, health literacy measurement research is growing exponentially and with this the discussion of its purpose, strengths, and limitations [24]. Some recommendations for the future are provided in the following.

First, health literacy has been measured for decades. The focus has changed over time, e.g., on literacy, information processing, navigation in health systems, and topic-related issues. Likewise, the tools are more and more sophisticated with strengths and limitations depending on their specific use [1,25]. The accumulated knowledge from health literacy measurement research combined with models and strategic thinking on data analytics and governance has the potential to inform the move away from lack of health literacy data to the generation of relevant health literacy analytics accommodating predictions for value-based, data-driven health-literate organizations.

Second, while advancing health literacy data generation from measurement to integrated monitoring mechanisms, it is recommended to lean on local contexts and participatory approaches that are strengths-based and solution-driven to promote health and equity [26]. The implementation of data analytics and predictions would not only inform and serve policy, research and practice in the organizations, but also promote the implementation of new tools and interventions to change the level of health literacy in an organization, community or setting.

Third, health literacy is a multi-dimensional concept that is content- and context-specific [26]. The tools to measure health literacy will, therefore, depend on the health literacy challenge that is explored and which aspects of health literacy are in focus as well the implications of the target group being researched. The data pipeline will need to be designed in ways that make use of both qualitative and quantitate data to embrace the scope and the scale of health literacy in all its facets. The health literacy dashboards that are being developed may not look the same as the choice of measurement tools and additional data generation will vary from organization to organization.

Fourth, health literacy measurement is instrumental in evaluating and monitoring development and progress at individual and population levels. Nonetheless, so far, much health literacy data are project-based and not part of any wider systematic health information system or data-driven approach. A well-constructed health literacy data pipeline that facilitates data collection, cleaning, integration, and explanation for strategic and supports business intelligence within health organizations and the wider eco-system, may help management to strategically identify trends and analyze data effectively at scale [22].

Fifth, from a start it is important to rally the need and identify the organizational and business value of health literacy analytics to secure the management buy-in, human resources and infrastructure to implement a successful data strategy that is future proof [6]. Moving from a focus on data and measurement generated from add-on projects towards built-in data analytics and generation of insights can help to maintain and promote the use of health literacy data to inform policy, research, and practice in the organization and beyond.

Sixth, health literacy data, in particular digital health literacy data, play a role in the digital transformation of health systems at large. The emphasis on digital health stresses the need for interoperability entailing the ability of different health literacy applications to access, exchange, integrate and cooperatively use data in a coordinated manner through shared application interfaces and standards, and within and across organizational, regional, and national boundaries, to provide seamless portability of information and optimize health outcomes [27].

Lastly, investing in health literacy data analytics and data governance can pave the way for the integration of health literacy as an acknowledged global health indicator in large-scale surveys, ventures, daily business and evidence-based policymaking. While the transformation towards value-based, data-driven health-literate organizations and systems may be under development; it can be accelerated with the engagement of a wide range of stakeholders and funding for the implementation of health literacy surveillance and monitoring. An inter-disciplinary approach is needed through the engagement of health literacy champions active in health, policy, academia, industry and civic society. Both the people served and the staff serving will benefit from the value created through health literacy analytics as a way to strengthen an organization’s vision, mission, and objectives to fulfill its purpose and for the public good of the wider society.

## Figures and Tables

**Figure 1 ijerph-19-13210-f001:**
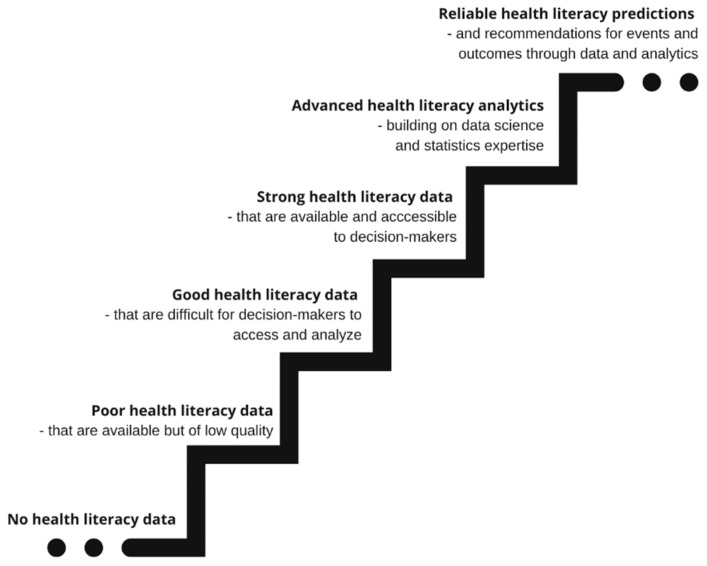
Progressing health literacy data and measurement to analytics and predictions.

## Data Availability

Not applicable.

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
