# Peer review of "From Project-Based Health Literacy Data and Measurement to an Integrated System of Analytics and Insights: Enhancing Data-Driven Value Creation in Health-Literate Organizations"

_ijerph, 2022, doi:10.3390/ijerph192013210_

Round 1

Reviewer 1 Report

Dear author,

The topic under discussion is very important and will have relevance across countries. Issues with data systems, governance, collection and utilisation are fairly universal, and we see this across almost every type of data in health.

As it currently stands, I feel the paper is too general and contains a lot of data jargon that the average reader may not understand. I think it can be strengthened by defining/describing the universal issues with data systems and governance, and then highlighting where the gaps/opportunities are for health literacy data specifically. I think the 'calls to action' also need to be clearer. If we want to improve data systems, who needs to step up and provide leadership in this space. Governments have a significant role, so do global health agencies, universities and other research institutions. They need to establish the foundations on which healthcare organisations can build and connect into. If the paper is able to articulate this more clearly, it is more likely to draw attention and hopefully motivate action.

Good luck with the paper going forward.

Note. I have made a comment and highlighted typos on the attachment.

Author Response

Dear reviewer

Thank you for the fruitful feedback. While the topic may seem broad and general in relation to health data at large, it is not yet the case with regards to health literacy in particular. 

Therefore, to stimulate reflection and action, the paper is written with the aim to build bridge between the fields of health literacy and data analytics. The purpose is to introduce concepts from both areas which may be new for some of the readers and perceived more general for others. Providing specific stepwise advice on how data analytics can be integrated is a way to help the reader get started with the implementation of a data-driven health literate organization. While much focus has been given to project-based academic measurement in the field of health literacy; less attention has been given to the implementation of a systemic approach to data analytics within organizations and companies which is the purpose of this introductory paper.

I have made an attempt to make the recommendations clearer in the conclusion and added a paragraph on who should be involved, in particular highlighting the need for an inter-disciplinary approach.

Typos were edited and the use of semi-colon reduced. Thanks for highlighting these issues.

Kind regards,

Kristine Sørensen

Reviewer 2 Report

An excellent summary analysis of how health literacy data could be included into an integrated system to be used for shaping health policy decisions. The degree of health literacy of the population is an important but often neglected addition to health related public policies. Knowing and taking this into account is essential for planning healthcare transformations, public health measures and even health impact assessments. The review contains many interesting insights, but perhaps it is also worth considering that politicians' understanding of health is also objectionable and needs to be changed.

Author Response

Dear reviewer.

Thank you for the encouraging words and for highlighting the role of politicians.

I have added points about evidence-based policymaking and the importance of engaging health literacy champions active in policy and other sectors.

Kind regards,

Kristine Sørensen

Reviewer 3 Report

Thank you very much for giving me the opportunity to revise this interesting manuscript by Sorensen K. on the need (and importance) of an integrated system of analytics and insights on health literacy data. 

The manuscript is well written and the topic is timely, as underlined by the Author, as the implementation of systemic health literacy data analytics is still scarce despite the growing amount of health literacy assessment research across the world. 

There is only one thing I recommend to the Author (and other minor issues), and it's to better specify how a whole new model of health literacy data analytics would be able to not only inform policy, research, and practice in the organizations, but also promote the implementation of new tools in order to change the level of health literacy in a community. I think this would be important for readers (especially those working in the field of public health) that know the topic and collaborate in projects related to health literacy and health literacy measurement and wonder which ideas and strategies to implement to improve its levels. 

I also suggest checking the use of semicolons in the text, and the lines 50-52 (something may be wrong with the sintax). 

Best regards, 

Author Response

Dear reviewer

Thank you for the valuable feedback.

I have edited the line 50-52 as noted and checked the manuscript for the use of semicolon.

Your point regarding the use of data analytics to inform new tools to promote health literacy in communities has been integrated. It was much appreciated.

Best regards,

Kristine Sørensen